# Artful Ageing, Not Just Successful Ageing

**Tine Fristrup** [1,*] **and Jon Dag Rasmussen** [2]

1    Danish School of Education, Aarhus University, 2400 Copenhagen, Denmark
2    Department of the Built Environment, Aalborg University, 2450 Copenhagen, Denmark
*    Correspondence: tifr@edu.au.dk

**Abstract:** In this article, we develop a tentative philosophy to orchestrate and support possibilities for artful ageing. This effort argues that older adults need a broader range of opportunities to explore the manifold ephemeral, non-rational, and in-between elements of an ageing life. The philosophy is rooted in the notion that older adults need space (literally and metaphorically) to explore and process their existence and that engaging in such processes can entail emancipatory effects in everyday life. The perspective unfolded throughout the article is a philosophical venture, or, rather, a preliminary work, developing the concept of artful ageing as a tool applicable in rethinking and broadening the range of activities occurring in institutional settings dedicated to older adults. Furthermore, the perspective also presents a critical stance towards normative footings and biopolitical agendas embedded in current regimes of active ageing. Artful ageing represents the ambition to enable and support artful lives, events, and activities among residents and participants in care homes and other contexts. We argue that physical and existential spaces are closely intertwined entities and that initiatives aimed at maintaining adequate measures of openness, ambiguity, and sensory intimacy, i.e., events that allow for the experience of metaphorical cracks, can afford artful pockets in which to reside for a while, seek refuge, recharge, stray from the beaten track, and obtain an always partial feeling of emancipation. In qualifying the concept of artful ageing, we hope to open new avenues to contemplate and subsequently initiate activities for older adults that are not just orientated towards physical health in later life. At the same time, our ambition is to develop a critical perspective aimed at challenging existing notions of successful ageing in (re)invigorating the importance of artful processes and experiences as an element inherent to successful ageing, as well.

**Keywords:** artful ageing; successful ageing; philosophy; emancipatory design

## 1. Introduction: The Model Ageing Enterprise

The dominant narratives on ageing revolve around the conceptual framework of "a new gerontology", engendering the concept of "successful aging" (SA) elaborated by John W. Rowe and Robert L. Kahn in 1998 [1]. This unfolding progress of gerontology was embedded in a positive approach toward ageing and effective functioning in later life. Breaking out of the disease framework in redefining "successful aging" can be traced back to their 1987 article in "Science" [2]. Rowe and Kahn stated in their book published in 1998 [1] that the goal was to move beyond the limited view of chronological age in retaining and enhancing people's ability to function in later life. In this context, the fear of ageing revolved around the narratives of disability as deprived functionality.

Rowe and Kahn's 1987 article [2] questions an understanding of the term "usual aging" as a naturalised ageing process evolving around faith or destiny, implying that it is not possible to intervene in and postpone the ageing process. The notion of natural ageing was challenged in the 1980s due to the introduction of the World Health Organization's "Ottawa Charter for Health Promotion" [3], which enabled a new perspective on health interventions in which people could increase control over and improve their health [3]. Because health was seen as a resource for everyday life and a positive concept emphasising social and personal resources and physical capacities, health promotion targeted the

individual's responsibility for their lifestyles and well-being, not just the health sector's accountability [3]. Higgs et al. [4] talk about "the will to health in later life" as a prerequisite in moving beyond the understanding of ageing as natural ageing and introduce the notion of normal ageing as the possibility to transgress the usual ageing term used by Rowe and Kahn [2]. In this way, health promotion [3] made the new gerontology possible, where the "usual" ageing process could be transgressed through a new vocabulary we now know as "successful aging". It is possible to be more or less "successful", which enables interventions around health in later life [4] through prevention programs.

During the late 1980s, prevention programs in different sectors encouraged older adults to engage in physical activities to postpone dependency, vulnerability, and disability in old age. Physical capacities are emphasised in well-being efforts, but well-being has more to it, as it is about enjoying life beyond exercising a healthy lifestyle. Both mental and social well-being have been brought to the table through the health promotion discourse, which has made it possible to approach later life as something beyond just being old and sick [4]. What Rowe and Kahn argued in their 1987 article [2] was that dichotomising the phenomenon of ageing into pathological versus normal states, respectively, did not capture the range of actual ageing experiences [2]. They differentiated "normal" ageing into "usual" ageing, in which individuals experience nonpathological age-related changes but are at high risk for disease, and "successful aging", in which nondiseased individuals experience high functioning and are at low risk for disease. This conceptualisation was nonetheless one-dimensional in its focus on objective physical functioning, a deficiency the authors wanted to address [5].

Rowe and Kahn's argument has given rise to a widespread discussion of notions of successful ageing. However, the term "successful aging" was first featured in an article by Havighurst in 1961 [6], where it was defined as "the conditions of individual and social life under which the individual person gets a maximum of satisfaction and happiness" (p. 8). Havighurst [6] contrasted activity theory, which construed successful ageing as the "maintenance as far and as long as possible of the activities and attitudes of middle age" (p. 8), with disengagement theory, which defines successful ageing as "the acceptance and the desire for a process of disengagement from active life" (p. 8). He advocated a subjectivist definition of successful ageing, emphasising well-being [5]. This emphasis was further elaborated in the Ottawa Charter for Health Promotion from 1986 [3].

In many ways, Rowe and Kahn continued Havighurst's efforts, even though they pointed towards a more objectivistic definition of successful ageing, building on the premise that specific characteristics, actions, and behaviours bring about successful ageing and that we can already know what these are. They stress that intrinsic lifestyle factors are influential [7] (p. 435). Definitions of successful ageing based on objective criteria define success by the judgement of others, omitting older adults' perceptions and experiences, thus focusing on functioning and the health state as measures of success, which puts many older adults in the "unsuccessful category" [5].

Interestingly, Rowe and Kahn provide little evidence or examples of reducing risk and enhancing resilience and recovery for parts of the older population at the most significant risk of ageing unsuccessfully [5]. The "unsuccessful" part is the least developed aspect of successful ageing mainly because the idea behind successful ageing is embedded in understanding the increase in older populations as a societal challenge—a problem to be addressed and solved. The notion of activity embedded in the understanding of successful ageing becomes the solution to the ageing problem in ageing societies [8]. According to Walker and Maltby [8]:

> "The problem with active ageing, like many scientific ideas that are transported into policy arenas, is that it lacks a precise universally accepted definition. As a result, it has quickly become common currency globally and, basically, all things to all people. The dominant policy paradigm across the globe is the economistic one of working longer (OECD, 1998, 2006). In contrast, the gerontological paradigm stretches back to research on 'successful ageing' and the connections between activity and health." (p. 2)

The political "translation" of the gerontological concept of successful ageing has resulted in a broadening of the concept with an emphasis on "active ageing" as an umbrella for words such as successful, productive, positive, and healthy ageing, or within the framing of Timonen [5]—"the model ageing enterprise" (p. xi).

## 2. The Aim: A Philosophical Inquiry

In this article, we develop the argument that we need to take a critical stance on active ageing ideation and examine the biopolitical efforts that make discourses on active ageing possible. In this regard, we also need to ask what the "active" part of this concept contains in contemporary gerontology and in the political appropriations of this notion circulating in our society. Furthermore, we will try to formulate a new approach to the understanding of active, successful, and healthy ageing as these notions arise when we approach them with perspectives from the arts and critical design. We do not offer a new ageing model, which would contradict our efforts debunking "the model ageing enterprise". Instead, we offer a philosophical inquiry into how an "artful" approach to ageing can expand our reflexivity regarding ageing as a process with existential, material, and emancipatory notions of what it is like to be "human" in later life. Rowe and Kahn [9] provided conceptual expansions in their concept on "successful aging" in favour of "successful aging 2.0", as they outlined the following argument: "To understand the complex relationship between aging at the societal and individual levels is perhaps the greatest gerontological challenge of our time" (p. 595).

Our aim in this article is to go beyond the dualism provided by "successful aging 2.0" [9] and the division between the social and the individual through poststructuralist and posthumanist theories provided by social scientists outside the field of gerontology. This will allow us to expand the emphasis of critical gerontology into modes of experimental encounters, which facilitates "ageing" as a process of living and learning "artfully". Artful ageing is not a model, nor does it have a clear definition; rather, it is a philosophical perspective that offers a basis for new discussions of ageing between existing and traditional scientific pillars in ageing research.

In conceptualising artful ageing, we came across G. D. Cohen's concept of "creative ageing" [10]. Cohen [10] emphasises that managing human potential is a journey in which discovering later life is an exciting prospect. He departs from the biology of ageing and focuses on factors that cause and contribute to the ageing process. In trying to unfold the creative mystery of biology, Cohen [10] demonstrates his subscription to the will to health in later life [4]. Empowerment strategies can reveal creative ageing to debunk the biomedical dominance, and the decline narrative can be exchanged through the conceptualisation of creative ageing for a narrative on resources, growth, experiences, and strength [11].

## 3. Governing Ageing by Regimes of Exercise

According to sociologist Nikolas Rose [12], empowering efforts evolving around health interventions in later life can be elaborated as "the politics of life itself", which has promoted "our growing capacities to control, manage, engineer, reshape, and modulate the very vital capacities of human beings as living creatures" (p. 3). Following these processes, social problems such as "ageing" become medicalised, and the making of biological citizens involves "the creation of persons with a certain kind of relation to themselves. Such citizens use biologically coloured languages to describe aspects of themselves or their identities, and to articulate their feelings of unhappiness, ailments, or predicaments" (p. 140). Furthermore, Rose [12] emphasises that the languages that shape citizens' self-understandings and self-techniques and are disseminated through "authoritative channels" (p. 141), i.e., health education, medical advice, books written by doctors about particular diseases, and television documentaries that chart individuals coping with particular conditions, promoting a medicalisation of social problems. This authoritative information interferes with the medicalisation of ageing in ageing societies, and preventive interventions evolve around new kinds of medicine, which paradoxically offer no medical solutions.

Arts on prescription (https://vbn.aau.dk/en/publications/an-arts-onprescription-programme-perspectives-of-the-cultural-ins (accessed on 11 June 2022)), i.e., culture on prescription, museums on prescription, dance on prescription, literature on prescription, and play on prescription, is a term used to describe interventions that use the arts to promote healthy behaviour in life in general and in later life in particular. The notion of well-being is addressed in the efforts of social prescribing, where health outcomes are linked to non-medical difficulties, such as deprivation, social isolation, housing, or unemployment, and involves connecting people to community groups/organisations to help with these difficulties [13]. Culture by Prescription is a Danish project in which four municipalities offered citizens with easy to moderate stress, anxiety, or depression participation in a 10-week group course with cultural activities two to three times a week. The project ran from 2016 to 2019; the evaluation was published in 2020 (https://www.sst.dk/da/puljer/kultur-paa-recept (accessed on 11 June 2022)). Museums on Prescription is a three-year research project (2014-17) funded by the Arts and Humanities Research Council that is investigating the value of heritage encounters in social prescribing (https://www.ucl.ac.uk (accessed on 11 June 2022)). National Alliance for Museums, Health & Wellbeing has a webpage containing material on well-being promoted by interventions offered through museums framed by a social prescribing approach (https://museumsandwellbeingalliance.wordpress.com (accessed on 11 June 2022)). A published review by the WHO [14] outlines the benefits of arts-based interventions as low-risk, highly cost-effective, and holistic options for complex problems, often with no medical solutions.

We can approach these types of interventions as embedded in a biopolitical agenda and implemented through what we could call bio-pedagogical interventions. "Biopolitics" is a term that refers to the intersection and mutual incorporation of life and politics. In literal terms, it signifies a form of politics that deals with life, i.e., bios in Greek. We can then discuss the biomedicalisation of ageing, which informs the biopolitics of ageing through the intersection of ageing life and ageing politics. Since ageing politics inform ageing life through the bifurcation of a gerontological agenda on successful ageing and a political agenda on active ageing—the ageing corporality is framed by the social as a political body, meaning that the body is not one's own—ageing is not entirely a private matter. We might even discuss "practiced bodies" [15] to overcome the dualism between the individual and the social elaborated in "successful aging 2.0". A disjoining of the individual and the society embraces an understanding of people being psychologically integral independently of social phenomena. Towards the end of the 20th century, social theorists began to view practice as the social phenomenon that constitutes individuals.

Michel Foucault is a prominent figure in the examination of the constitution of modern persons and subjects by applying disciplinary techniques and scientific discourses to the human body. Foucault's views are widely disseminated, and include his development of bio-disciplinary power, where the body (and self) is produced through modern techniques and discourses [16]. The particular ways of being a subject are constituted by applying techniques and discourses to human bodies.

The socially invested body is a material thing with a structure, organs, functions, sensations, pleasures, and physiological processes and systems in which biology and history are bound together through practices of bio-disciplinary power [15]. Practices fashion the person and subjecthood by drawing out the features of bodily existence through the shaping of attention to, perception of, and thought about oneself embedded in possible conceptualisations within discourses. With a Foucauldian approach, we can debunk "functionalist gerontology" as the gerontological discourse on successful ageing, primarily focused on older people as a social problem and lacking conceptual tools that impinge on power, knowledge, surveillance, discourse, governmentality, and technologies of the self [17].

Applying a Foucauldian approach has implications for how ageing can be understood as a discipline of study and a social process [18]. When interrogating how knowledge in the modern era has been organised and legitimated [16], we can interrogate the complex

interactions between current social policy, popular culture, institutions, and older people [18]. We refer to the discourses, perceptions, sites, and practices that are conditions for the emergence of gerontological knowledge [17].

In this case, we examine how ageing has been organised, shaped, and positioned through knowledge systems and social processes, which impinge on the social construction of ageing, even though Foucault had nothing to say about ageing. According to Powell and Wahidin [17]:

> *"Aging becomes governed by regimes of exercise which individuals become the object of their own gaze in order to maintain their commitment to training regime to achieving a particular body project generated through discourses from science. Bio-medicine may make people 'healthier' and 'live longer', but they are still not freer people from the structures imposed upon them within their society. [ . . . ] However, bio-medicine still struggles with the notion that being old is positive, in relation to the ideology of aging especially old age as 'decrepit' from decades of negatively stereotyping senescence. These structures change within bio-medicine in order to reconstruct narratives of aging, but their dominant discourse of decline is still their master narrative of legitimacy."* (p. 12)

The master narrative of ageing as decline frames the biomedical interventions we witness in today's ageing societies. For individuals to survive within the contemporary ageing society, the argument espoused is that they must change their attitudes to enjoy the benefits of active ageing professed by biomedical staff as in the social prescribing because ageing needs modification as though it is a medical problem to begin with.

The master narrative of ageing is framed by understandings of degeneration, which Carroll L. Estes has critically embraced in her book on "the aging enterprise" [19], and an article on "the biomedicalization of aging" [20]. Social environments in which people live do not influence the biological and psychological views on the lives of older adults. These dominant knowledge perspectives on ageing remain reproduced through biopolitical interventions in not just older people's lives, but throughout life framed by a life course perspective following the WHO's framework for active ageing in 2002 [21].

Even though understanding of the ageing process is rooted in biomedical responses, we can approach ageing from a Foucauldian perspective and examine how ageing is regulated, classified, and shaped by social discourses of Western culture [22]. This approach enables the scope of ageing to be broadened beyond biomedical accounts of the body following perspectives on the posthuman condition.

## 4. Parts of Ageing Lives between Bios and Zoe

Posthumanist Rosi Braidotti [23] proposed "the primacy of life as zoe":

> *"I oppose zoe, as vitalistic, prehuman, generative life, to bios, as a discursive and political discourse about life. I want to defend the argument that the emergence of these discursive "bits of life" results in the need for more social and intellectual creativity in the scientific as well as the mainstream culture."* (p. 177)

The social prescribing of arrangements dedicated to older adults in the arts is often approached as a preventive intervention following health promotion. Following Braidotti [23], bios engender the political discourse about life as bios and become the preventive efforts for successful ageing. In other words, successful ageing originates in "the logics of bios" and in the distribution of current biopolitical understandings regarding a healthy, active, and, thus, successful elderly human being. Meanwhile, this success also requires acceptance, internalisation, and the subsequent performance of adequate ageing.

Older adults will therefore have to express and perform their age according to certain normative and functional ideals, a process potentially leading towards both "adequate" and "inadequate" ageing, or rather, to maintain the terminology applied above, towards successful and unsuccessful ageing. In this regard, we need to ask the following question: How do we age inadequately and unsuccessfully? Moreover, simultaneously, we will have to engage in an explorative venture led by the following question: Where and how do we

provide space for both (so-called) inadequate and unsuccessful ageing? Furthermore, we argue that we (as a society and a welfare state) are obliged to offer possibilities for an artful, thus nuanced, open and rich old age and that this ambition requires rethinking how old age is constructed and staged.

We opt for including zoe as a prerequisite in making possible new engagements in older adults' lives, hopes, desires, and contributions. The artful engagements can, to some extent, problematise views of older adults dominated by medicalised, individualised, and pathologised accounts. We build on the notion that physical and existential space are closely intertwined in the human lifeworld and that the presence of and exposure to art and artistic processes, therefore, possess both educational and emancipatory effects in the lives of human beings. Our point of departure is that life is immanently carried out (and performed) in architectural and designed spaces; thus, to exist is to exist in and among designs of various kinds.

## 5. Ageing Materialities

Following Elizabeth Grosz's conceptualisation of the body as "open materiality" [24], we cannot approach the body as either a culturally inscribed product of the social or as simply part of biology/nature. Instead, the body can be approached as open materiality existing as a borderline entity between the binary poles of the nature/culture dichotomy [24]. According to Grosz [25]:

> *"Joy, affirmation, pleasure, these are not obstacles to our self-understanding, they are forms of self-understanding. And if life is more and more oppressive, then in a way it is only these small pockets of knowledge production, art production that provide a counter to the weight and emptiness of everyday life. So we need to affirm, we need a place where we can simply affirm. The rest of the world is bleak enough. I mean the point is the way in which the new world is produced is precisely through revelling in the affirmation of the strengths that art gives us. The only way we can make a new world is by having a new horizon. And this is something that art can give us: a new world, a new body, a people to come."* (p. 256)

As Grosz [25] proposes, pockets of artful, explorative, and potentially experimental experiences can challenge the increasingly burdensome elements of old adulthood, the physical and social conditions that inevitably accompany older adults' everyday life. She offers possibilities for thinking about artful processes, a concept we would rather apply than the production concept employed by Grosz [25], as a means to joy, affirmation, and pleasure, and as a way in which to remedy the bleakness of certain parts of everyday life.

Linn Sandberg [26] argues for the need for theorising old age beyond the binaries of decline and success. She draws on the work of the abovementioned feminist corpomaterialists Rosi Braidotti and Elisabeth Grosz. Sandberg's article proposes affirmative old age as an alternative conceptualisation of old age [26]:

> *"The decline narrative, on the one hand, is highly centered on the decline of the ageing body as frail, leaky and unbounded, and on how old age is characterised by non-productivity, increasing passivity and dependency. Discourses on successful ageing, on the other hand, rely heavily on neo-liberal imperatives of activity, autonomy and responsibility. In successful ageing, the specificities of ageing bodies are largely overlooked while the capacity of the old person to retain a youthful body, for example, with the aid of sexuopharmaceuticals, is celebrated."* (p. 11)

Sandberg [26] argues that there is a need for terminology and language regarding old age that goes beyond the binaries of decline and success, as well as the Cartesian body and mind dualism. According to Sandberg [26], "the material body should be understood as possessing force and agency to also shape subjectivity and sociality, and not merely as malleable raw material taking shape in socio-cultural discursive regimes" (p. 17).

The material turn in the social sciences, cultural studies, and the humanities has reached the field of gerontology as a new material gerontology, according to Gritt Höppner

and Monika Urban [27], and their examination of "Materialities of Age and Ageing", which "comprises both the fleshy-sensual experiences of human bodies and their interplay with and relation to non-humans, such as commodity items, things, technologies, architecture, and spaces" (p. 5). Through a material approach to ageing, they want "to relieve human subjects of some of the burden of an inexorable ageing process; thus, it relativizes the responsibility ascribed to seniors to perform a "successful" and therewith delayed ageing process" (p. 5). The perspective offers a scepticism towards the idea that ageing takes place solely *in* humans and is aligned with both cultural gerontology and critical gerontology [27]:

> *"Addressing these materialities of ageing brings up its own critique on definitions of ageing bodies and material environments. This framing does not presume that age and ageing are solely products of human-to-human interactions or those of formative environments or of discourses. Rather humans, non-humans, and discourses become essential parts of ageing processes. Such a material framing enables us new insights into forms of age and ageing and thus offers an opportunity for scholars to engage critically with materialities of age and ageing."* (p. 3)

Posthuman theorisations and the material turn reflects how the human condition has become decentred and hybridised, which anticipate contemporary and future mixes of what it means to live in what we call spaces of ageing. The spatialisation of ageing can be understood as "the process of causing something to occupy space or assume some of the properties of space" (https://www.collinsdictionary.com (accessed on 13 January 2023)). When older adults occupy, inhabit, and form parts of space, they must claim the properties of that particular environment. Different forms of design and architectural spaces are always loaded with intentions and motives and can be understood as interpretations and manifestations of past and current views on humanity. The spaces of ageing often encountered and inhabited by older adults impose certain normative ideals and understandings of these "users".

Meanwhile, and as the posthuman and material theorisations support and engender, an emancipation from the medicalisation, rationalisation, and incarceration inherent to the social spatiality of ageing can occur. Such an emancipatory effort can be approached by combining the concept of emancipatory design with the preliminary conceptual framework of artful ageing. As we argue in the following, this perspective can be part of redeeming and reconceptualising spaces of ageing as spaces of artful ageing.

## 6. Emancipatory Design and Spaces of Artful Ageing

Building on our critique of the model ageing enterprise in general and the paradigm of successful ageing in particular, we find that the concept of success must be nuanced and supplemented or rebooted and reconstructed. In this ambition to disrupt the biopolitical foundation on which successful ageing is based, the first critical question asks what the actual definition of success could be. How do we measure the success inherent to successful ageing? What, if not health and capability to self-care, constitutes a viable basis for assessing, describing, and comparing success in later life?

As the sections above seek to highlight, the downsides of successful ageing in the current understanding and definitions are manifold because they leave ageing adults with a very sparse scope to manoeuvre and construct ageing practices and lives, which are acknowledged and recognised within this paradigm. In other words, the prevailing ideals concerning successful ageing do not celebrate actions and elements of seemingly artistic, aesthetic, irrational, open-ended, and even spiritual character.

Following Braidotti's proposal to introduce life as zoe as a vitalistic perspective, we have found a practical approach to challenge and reconstruct the notion of successful ageing, thus paving the way to novel definitions. Meanwhile, this active and maybe even subversive act demands that another element be added to the equation, namely, the element of artfulness.

We argue that practices of artful ageing (tentatively defined below) are essential components in the everyday lives of many older adults, thus contributing to a holistic experience

of meaningfulness, gratification, and well-being (occurring from, e.g., sensory stimulation). Moreover, by introducing the vitalism characteristic of life as zoe, we re-discover a range of core human needs that are not represented, and therefore not distributed nor promoted, within the existing ideals of successfulness related to ageing. With the concept of artful ageing, we seek to broaden the scope of practices, activities, and orientations that can be deemed both legitimate, appropriate, and advisable in the everyday lives of older adults. Furthermore, by extension, we aim to formulate a concept that can contribute to creating metaphorical cracks that hold emancipatory potentials and effects in older adults' everyday lives.

At this point, it is necessary to define what we mean by the term artful, which constitutes the first part of artful ageing. Such a definition is tentative, and must therefore be understood as preliminary, open, and changeable throughout the article (and beyond).

Merriam Webster's online dictionary has three definitions of artful: (1) something that is performed with or showing art or skill; (2) something that is characterised by art and skill; and (3) something artificial, i.e., made by human hands (https://www.merriam-webster.com (accessed on 5 January 2023)). Artful ageing directly refers to something akin to knowledgeable and skilled ageing, thus implying the actual ageing process. To age artfully is to engage in the ageing process with arts and skills and perform artful ageing or indulge in the ageing process in artful manners. Artful ageing is actively shaping and fashioning the ageing process and old age. Furthermore, being artful can imply the presence of cleverness and specific skills allowing someone to achieve what they want. We add another meaning to these understandings of artful, concerned with artistic processes and forms of expression, recognition, and experience. Here, artful implies the engagement with art in a more conventional use of the term.

What emerges from these definitory exercises is the contours of a concept apt to describe and frame experiences in later life that are characterised by skillful, knowledgeable, and artistic elements, emphasising the latter. We argue that artistic endeavours bear their entitlement and that the ageing process depends on experiences that allow for the broadly defined spaces occurring when engaging in open-ended, experimental, and aesthetic processes and events. According to activist philosopher Erin Manning [28], artful practices honour complex forms of knowing, and they "[...] are collective not because they are operated upon by several people, but because they make apparent, in the way they come to a problem, that knowledge at its core is collective" (p.13). Here, Manning [28] acknowledges the social and spatial dimensions adhering to artful and artistic practices, reminding us that they are immanently situated in physical and social space and always involve others.

In working with the concept of artful ageing, we depart from the notion that artful experiences arise in different types of cooperation and entanglement between someone and something, i.e., of a profoundly collective character. On a more concrete level, this means that artful ageing depends on both social and physical space, or rather, on a socio-physical space allowing for these processes and events to occur.

At this point, artful ageing is tentatively defined as a perspective arguing for the necessity of artful experiences in later life. In qualifying the philosophical framework for artful ageing, we draw on the concept of emancipatory design (ED) [29] as a perspective highly attentive to the inevitable question of space in this regard. In working with the general observation that physical elements make up disparate obstructions to different people, and with the acknowledgement that personal as well as socio-cultural circumstances directly influence these encounters between people and their environment, ED insists on taking the particular factors, the very concrete physical and socio-cultural contexts, its people, and singular characteristics into account when working with human–environment relations. ED offers a perspective suited to contemplate and analyse the barriers and obstacles posed by design and architecture, but more importantly, to operationalise the notion that design can be a means of (always partial and situational) emancipation, and that design holds emancipatory potentials. In this regard, ED is retrospectively oriented, as well as future oriented. What ED contributes to the discussion on artful ageing is

that artfulness requires a context and a situation. The attempts to provide experiences of artfulness to others, i.e., to older adults in this case, requires both physical and socio-material orchestrations, which can be understood as "design" in an extended definition, providing appropriate and valuable "pockets" [30] for artful processes in everyday life.

We find that the combination is mutually generative when engaging with artful ageing and emancipatory design. Whereas processes of artful ageing require both physical and socio-existential space (a concrete physical location with associated coordinates, along with an event affording certain socio-existential qualities), emancipatory design is defined by the ambition to liberate people from socio-material barriers and constraints encountered in everyday life [29]. When asking the fundamental question adhering to ambitions of emancipatory design: "Emancipation from what (?)", we can answer from the position of artful ageing, leaning on the critiques of the successful ageing paradigm developed above: Emancipation from agendas and narratives of ageing rooted in bio-political discourses; emancipation from incitements towards successful ageing that does not acknowledge a need for artistic, aesthetic, irrational, and open-ended excesses in everyday life; emancipation from barriers impeding the possibilities for encountering artful experiences; emancipation from the model ageing enterprise.

Moreover, we can depart from Braidotti's [23] vitalism in launching artful ageing as a tentative, yet critical philosophy to secure and enable possibilities for artfulness in later life arenas. Embracing life as zoe thus pointing towards a new and inclusive paradigm of successful ageing, and calls for a rethinking and a redefinition of this powerful term.

### 7. Concluding Remarks: Moving beyond the Binary

When framing "success" through the gaze of "artful ageing", we embrace a spacing of ageing, which emancipates the different interpretations of "ageing", enabling older adults to establish small cracks in their everyday lives, making life more livable, more moveable, and, hopefully, more enjoyable. We talk about a bodily presence in space, where older adults' appearance consists less in their material presence than in their interwoven connections, which comprise the ageing spatiality. Artful ageing can be practised in concrete places where encounters between people occur and where they enjoy the qualities inherent to what Herman Hertzberger [31] described as places where different age groups meet or have contact, where people interact, and unexpected (not planned) things take place more or less spontaneously.

Places where ageing is felt through "relationscapes" [32] address relations in movement more generally and lend a certain ubiquity to movement, which Manning [32] defines broadly to include everyday movement, movement of thought, scientific experiments measuring motion, paintings that map movement, choreographed body movement, and choreography as it is represented in film. Through relationscapes, we can "performatively bridge the gap between thinking/speaking and doing/creating" (pp. 1–2) through "matrices of cultural becoming" (p. 2).

Following Margaret Morganroth Gullette [33], we are aged by culture, and society constructs age anxiety in which ""temporality", from the vocabulary of an earlier paradigm, cannot reckon with life time at its most grippingly personal: the Aging *Me*. Many transformations of selfhood make better sense if studied through the everyday mediations of age culture" (p. 29). Our contribution to this article is a gentle disturbance of the everyday mediations of age culture.

We opt for more "tuned" spaces of ageing, where their attunement presents them as expansive spaces, in which atmospheres are experienced through immersion, underlining how they affect our dispositions. Following Gernot Böhme's [34] elaborations on "atmospheric architecture", we provide a horizon in which things and people appear as their lives play out. In this horizon, "ageing" becomes an atmosphere, and artful ageing becomes a "tissue of space and time", to paraphrase Walter Benjamin [35]. The tissue's elasticity depends on the moveability in the spatial arrangements, which makes new engagements possible in later life.

Herman Hertzberger [31] states that the habitable in-between becomes inviting spaces. The question is whether we can approach "ageing" as something "in-between" the official and dominant discourses on ageing (between success and decline/failure; natural and normal; active and passive; positive and negative; productive and unproductive, etc.) that could become habitable and unfold as something open for becoming significant through spaces of artful ageing, thus reminding us of the necessary space in which to exist.

What kind of spaces do we envision when we talk about existential spaces? Does this notion include anywhere in which existence happens and occurs? Are existential spaces inviting and apparent, spaces in-between as "the small cracks of everyday life" [36], or are they both and beyond? Furthermore, what is the relation between the notions of existential spaces (plural) and existential space (singular)? Suppose the former addresses a physical and material world. In that case, the latter concerns the immaterial and social dimensions inherent to everyday human life—the sphere described with the core phenomenological concept of the lifeworld. Is it as straightforward, dualistic, and definable as that, or are existential spaces something which are far more blurry, nebulous, imprecise, and mixed up; something not yet determined, undecided, and open for actions and attribution of significance—just waiting to be articulated, crafted, and formed where lives are being lived, and in the practice of everyday life?

Regardless of these distinctions, we live lives, we do lives, we perform lives, and we name the lives we live—something. We might live momentarily, instantly connect, make contact, resonate, operate, perform, and sculpt our lives as we live them. Afterwards, and often retrospectively, we name our lives' particular "events" by adding words and furnishing them through articulations.

When bodies meet in spaces, they become specific types of bodies through naming practices, often followed by shaming and blaming the bodies because they could have been or should have been different, according to societal norms.

We live by our bodies in spaces already defined and occupied with words through processes of articulation embedded in "the ageing model enterprise". They do not invite anymore—they lose their ability to be inviting spaces when becoming identifiable spaces. The bodies lose spacing, and the moving between other bodies is already established as specific forms of movement between already identified bodies.

The material and the discursive intertwine through possible and potential in-betweens through myriads of spatial, temporal, and minor gestures, as well as through spaces that invite people of all ages to connect differently, could engender inviting spaces as unexpected and emerging encounters in-between different people, enabling more liveable processes of ageing, where "successes" are shaped beyond the binary of "success and failure".

**Author Contributions:** This article results from a collaborative research process involving the two authors' mutual development of a novel philosophical conceptualisation of artful ageing. The first author has facilitated and led the writing process. All authors have read and agreed to the published version of the manuscript.

**Funding:** This research received no external funding.

**Institutional Review Board Statement:** Not applicable.

**Informed Consent Statement:** Not applicable.

**Data Availability Statement:** Data sharing not applicable.

**Conflicts of Interest:** The authors declare no conflict of interest.

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
