# Peer review of "Artful Ageing, Not Just Successful Ageing"

_2673-9259, doi:10.3390/jal3020014_

Round 1
Reviewer 1 Report
I recommend shortening the many long quotes, or simply paraphrasing them. Especially, I'd recommend not beginning with a long weird one that made me want to stop reading immediately. As I see it, they don't contribute anything and end up standing in the way of your argument, which only begins to take on the contours of something I can understand once we reach section 7 and 8. As I see it, all the build-up description of 'successful ageing' and bio-power etc. can be severely reduced and compressed and to a large extent taken for granted. My main issue with the article is a tendency towards using too many concepts, writing in a too convoluted style bordering on obscure, and referencing too many theorists instead of getting faster to the point of what 'artful ageing' may be and why it's important to conceptualize prior to, perhaps, studying it empirically. I may be coming from a very different tradition, I do, but I feel pretty confident that simplifying the writing, stating the argument and answering the 'so what' question all good academic writing should do very quickly, would benefit this important work on theoretical conceptualization, which is very interesting.
Author Response
We have improved the writing of the article on all of the reviewer’s accounts:
The argumentation is made more precise and consistent following previous and present theoretical background into a philosophical inquiry as the article's main aim. The research design is further elaborated through a more consistent argumentation, enabling the article to raise new research questions and bring a philosophical inquiry into developing a new framework for artful ageing. This has improved the overall argumentation of the article, which has become more coherent, balanced and compelling. We have shortened most of the long quotes by paraphrasing them. We have changed the content of the abstract for it to grasp the article's essence. In doing so, we have rewritten the whole article – especially the last first part of the article is replaced by a new introduction following a paragraph on the article's aim. We have reduced several of the statements according to the reviewer’s suggestions regarding the fundamental argument in the article. We have also reduced the complexity of the many concepts used to unfold the conceptualisation of what artful ageing is and why it is essential. Overall, we have simplified the writing and stated the argument on the theoretical conceptualisation, which the reviewer finds very interesting.
Reviewer 2 Report
Please, find my comments attached.

Author Response
We have improved the writing of the article on all of the reviewer’s accounts:
The parts on emancipatory design and artful ageing are developed further, and the many concepts are reduced and focused on in the second part of the article, which is now balanced following the first part when introducing “successful aging”. In this case, we have tried to answer all the questions raised by the reviewer in the first part of the review. We have unfolded the relationship between Cohen’s conceptualisation of “creative ageing” and our framework on artful ageing, as requested by the reviewer. One of the authors has already written articles on this topic, so it was easy to elaborate on the differences between Cohen’s and our approach. Instead of empirical examples, we have elaborated the argumentation by exemplifying the points taken in the article. We have introduced our research questions at the beginning of the article, and the abstract is rewritten following an emphasis on stating the aim of the article and the philosophical inquiry more strongly. The beginning of the article is rewritten and starts with an introduction, and the aim is to make our efforts more apparent to the reader.
Round 2
Reviewer 1 Report
The paper now makes a lot more sense and I'm happy to recommend publication and hope the notion of 'artful ageing' will gain even more traction!
Author Response
Thank you to the reviewer for recommending publication.
Reviewer 2 Report
The revised draft is a much better read compared to the previous one because the structure of the article is clearer and the different argumentative strands are unfolded in a more systematic and explicit way. In my opinion, this article as is a welcome addition to the scholarly debate on how to move beyond the success/decline binary that structures our thinking about older age (and consequently influences how older age is lived and experienced). The philosophical exploration of artful aging will provoke further thought on the topic.
Some minor suggestions:
p. 2, line 200 and line 236: I do not understand the significance of specifically the Ottawa Charter for Health Promotion for the argumentation.
p. 2, lines 205-225: The distinction between normal and natural aging and connection to “will to health” is still a bit confusing.
p. 3, line 318-319: the last part (“even though…. beyond the understanding of biological ageing” seems contradictory compared to what you have written about Cohen’s creative aging before.
p. 6, lines 690-697: The jump from “artful” to “art” and from “art” to “architectural and designed spaces” goes a bit too quickly in my view (I am a scholar in aging and the arts). This is a more general comment as well. I think you could prepare the reader even better here for what you will start arguing under section 6 on emancipatory design.
Author Response
Some minor suggestions:
- 2, line 200 and line 236: I do not understand the significance of specifically the Ottawa Charter for Health Promotion for the argumentation.
- 2, lines 205-225: The distinction between normal and natural aging and connection to “will to health” is still a bit confusing.
The section is now rewritten for clarity:
Rowe and Kahn’s 1987 article questions an understanding of the term usual ageing as a naturalised ageing process evolving around faith or destiny, implying that it is not possible to intervene in and postpone the ageing process. The notion of natural ageing was challenged in the 1980s due to the introduction of the World Health Organization’s Ottawa Charter for Health Promotion (WHO, 1986), which enabled a new perspective on health interventions, where people could increase control over and improve their health (WHO, 1986). Because health was seen as a resource for everyday life and a positive concept emphasising social and personal resources and physical capacities, health promotion targeted the individual’s responsibility for their lifestyles and well-being, not just the health sector's accountability (WHO, 1986). Higgs et al. (2009) talks about “the will to health in later life” as a prerequisite in moving beyond the understanding of ageing as natural ageing and introduce the notion of normal ageing as the possibility to transgress the usual ageing term used by Rowe and Kahn (1987). In this way, health promotion (WHO, 1986) made the new gerontology possible, where the “usual” ageing process could be transgressed through a new vocabulary we now know as “successful ageing”. It is possible to be more or less “successful”, which enables interventions around health in later life (Higgs et al., 2009) through prevention programs.
- 3, line 318-319: the last part (“even though…. beyond the understanding of biological ageing” seems contradictory compared to what you have written about Cohen’s creative aging before.
This paragraph has been deleted:
In this case, we follow Cohen’s conceptualisation of creative ageing (2000), even though we outline artful ageing beyond the understanding of biological ageing.
- 6, lines 690-697: The jump from “artful” to “art” and from “art” to “architectural and designed spaces” goes a bit too quickly in my view (I am a scholar in aging and the arts). This is a more general comment as well. I think you could prepare the reader even better here for what you will start arguing under section 6 on emancipatory design.
The two sections has been rewritten for clarity with an emphasis on linking artful ageing to “a context and a situation”, which is elaborated with an emphasis on “both physical and socio-material orchestrations”, and understood as “design” – so ED is a conceptualisation of the way we contextualise artful ageing in relation to “space and place” following physical and socio-material orchestrations:
In working with the concept of artful ageing, we depart from the notion that artful experiences arise in different kinds of cooperation and entanglement between someone and something, i.e. of a profoundly collective character. On a more concrete level, this means that artful ageing depends on both social and physical space, or rather, on a socio-physical space allowing for these processes and events to occur.
At this point, artful ageing is tentatively defined as a perspective arguing for the necessity of artful experiences in later life. In qualifying the philosophical framework for artful ageing, we draw on the concept of Emancipatory Design (ED) (Rasmussen & Torkildsby, 2022) as a perspective highly attentive to the inevitable question of space in this regard. In working with the general observation that physical elements make up disparate obstructions to different people, and in the acknowledgement that personal as well as socio-cultural circumstances directly influence these encounters between people and their environment, ED insists on taking the particular factors, the very concrete physical and socio-cultural contexts, its people and singular characteristics, into account when working with human-environment relations. ED offers a perspective suited to contemplate and analyse the barriers and obstacles posed by design and architecture, but, more importantly, to operationalise the notion that design can be a means of (always partial and situational) emancipation, that design holds emancipatory potentials. In this regard, ED is retrospectively oriented as well as future-oriented. What ED contributes to in the discussion on artful ageing is that artfulness requires a context and a situation. The attempts to provide experiences of artfulness to others, i.e. to older adults in this case, requires both physical and socio-material orchestrations, which can be understood as “design” in an extended definition, providing appropriate and valuable “pockets” (Rasmussen & Dannesboe, 2021) for artful processes in everyday life.